# A Rare Complication of Ascariasis: A Case of Acute Interstitial Nephritis

**DOI:** 10.3390/diagnostics13122054

**Published:** 2023-06-14

**Authors:** Nazareno Carullo, Francesca Divenuto, Nadia Marascio, Neill James Adams, Aida Giancotti, Nicolino Comi, Teresa Faga, Davide Bolignano, Giuseppe Coppolino, Francesca Serapide, Chiara Costa, Carlo Torti, Giovanni Matera, Angela Quirino, Michele Andreucci

**Affiliations:** 1Nephrology and Dialysis Unit, “Magna Graecia” University of Catanzaro, 88100 Catanzaro, Italy; 2Clinical Microbiology Unit, Department of Health Sciences, “Magna Graecia” University of Catanzaro, 88100 Catanzaro, Italy; 3Infectious and Tropical Disease Unit, “Magna Graecia” University of Catanzaro, 88100 Catanzaro, Italy

**Keywords:** acute interstitial nephritis, acute kidney injury, ascariasis, helminthic parasites, fertilised egg, renal failure

## Abstract

Acute interstitial nephritis (AIN) due to helminths is a rare cause of acute kidney injury (AKI). Helminthiases often progresses insidiously, making diagnosis difficult. This was the case of a 72-year-old man, who presented with renal failure, itching and diarrhoea. Urinalysis revealed leukocyturia, microhaematuria and mild proteinuria. A full blood count revealed leucocytosis with eosinophilia. A stool parasitological examination revealed fertilised eggs of *Ascaris lumbricoides.* Tubulointerstitial nephropathy secondary to *A. lumbricoides* infection was suspected. A percutaneous renal biopsy was not performed since the patient refused the anti-platelet therapy discontinuation. Mebendazole, albendazole and prednisone therapy was administered. After worm eradiation and discharge, recovery from the parasitosis, absence of pruritus and eosinophilia, and progressive improvement of renal function were observed, strongly suggesting a causal relationship between *Ascaris* infection and AIN. Parasite infection should be considered in the differential diagnosis of unexplained renal failure because early diagnosis and treatment are necessary to avoid irreversible complications.

**Figure 1 diagnostics-13-02054-f001:**
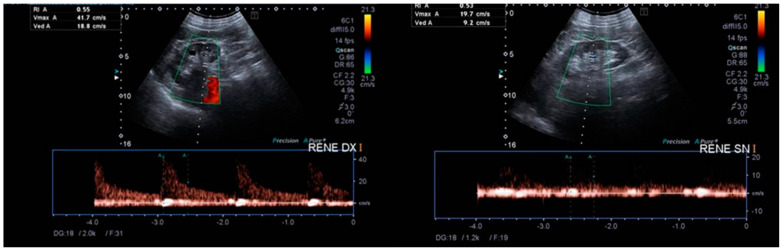
Renal ultrasound with echo-colour Doppler of the intraparenchymal renal vessels (green boxes). Kidneys were in place, within normal size limits, both with regular cortical thickness and echogenicity, with no obvious stones or signs of urostasis. On ECD, no velocimetry tracing changes were appreciated in the intraparenchymal arterial vessels explored, with IR values of 0.55. Due to kidney failure, a 72-year-old man was admitted to the Nephrology–Dialysis Operating Unit of the “Magna Graecia” University Hospital of Catanzaro, Italy. He presented with diffuse itching, loose stools and a recent admission to a local hospital for bilateral lower back pain and sudden worsening of renal function. The bilateral lower back pain was initially attributed to the expulsion of urinary micro-calculi while diarrhoea was ascribed to a common gastroenteritis. Due to a suspected allergic reaction to the drugs, amlodipine and omega-3 fatty acids were alternatively discontinued without any benefit. He had no history of use of antibiotics or non-steroidal anti-inflammatory drugs (NSAIDs). A few months earlier, both serum creatinine levels and urine tests showed normal values. This excluded the possibility that renal function was already (chronically) impaired. The patient was apyretic, with no cardiac, pulmonary or abdominal problems and no oedema. He presented diffuse scratching lesions on his limbs and back that had been present for the past several weeks. Laboratory parameters confirmed an impaired renal function (serum creatinine 3.94 mg/dL; eGFR (CDK-EPI) 14 mL/min/1.73^2^; blood urea nitrogen (BUN) 64 mg/dL), while urinalysis showed microhaematuria, leukocyturia and mild proteinuria (<1 g/day). Urine culture was negative. A full blood count revealed eosinophilia (absolute number of eosinophils 3616/µL; 29.4% of total leukocytes) and mild normochromic anaemia (Hb 11.5 g/dL). The autoantibody panel, viral markers, C3, C4, serum immunoglobulins, serum protein electrophoresis and light-chain proteinuria provided negative results. Abdominal ultrasonography showed both kidneys within normal size and regular cortical thickness with a slight increased echogenicity. Colour Doppler Ultrasound of intraparenchymal renal arterial vessels was also normal (resistance index (RI) 0.55) (Figure 1).

**Figure 2 diagnostics-13-02054-f002:**
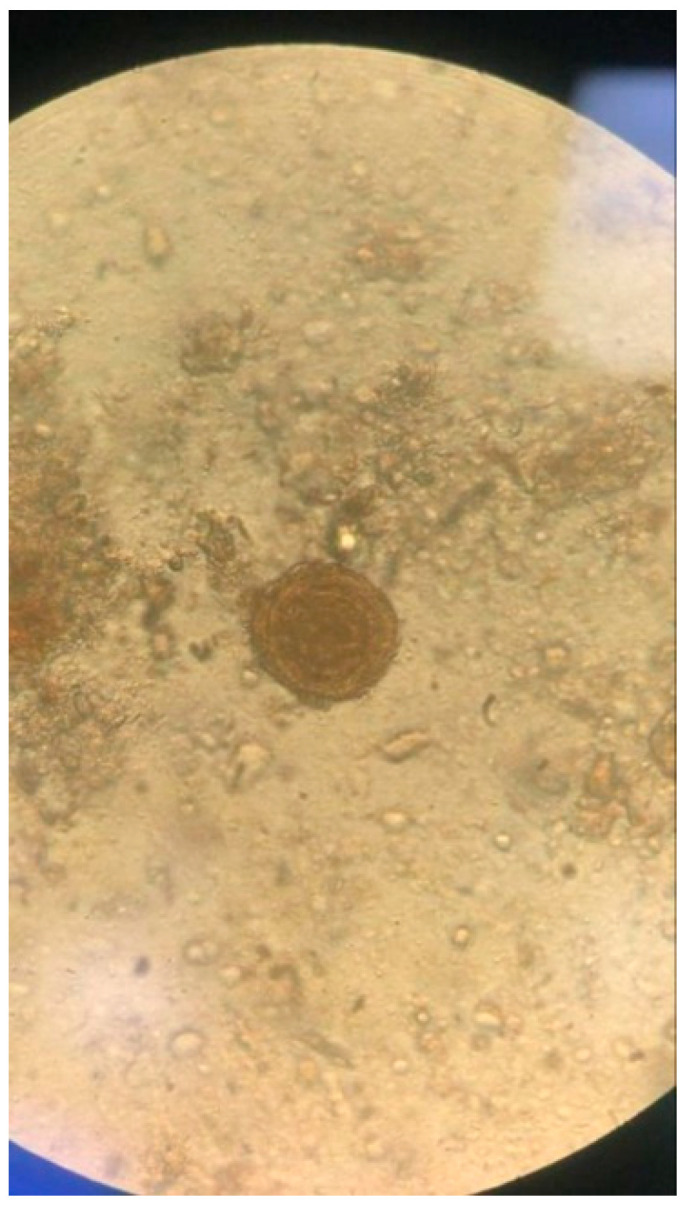
Fertilised egg of *Ascaris lumbricoides* in the faecal sample (400×). The fertilised eggs of Ascaris are rounded, with a diameter of between 55 and 75 μm and a width of 35–60 μm. They have a golden yellow to brown colour. They are characterised by a thick shell with a mamelon layer in the outer portion. Following the Ridley method, the stool specimen was treated by using formalin plus ether and pellet obtained after centrifugation was used for the microscopic exam. Potential allergic, parasitic and haematological causes of the eosinophilia were investigated. Serum IgE (PRIST) levels (571.1 kU/L) were increased and stool parasitological examination revealed fertilised eggs of *Ascaris lumbricoides* (Figure 2) after Ridley enrichment of the stool sample.

**Figure 3 diagnostics-13-02054-f003:**
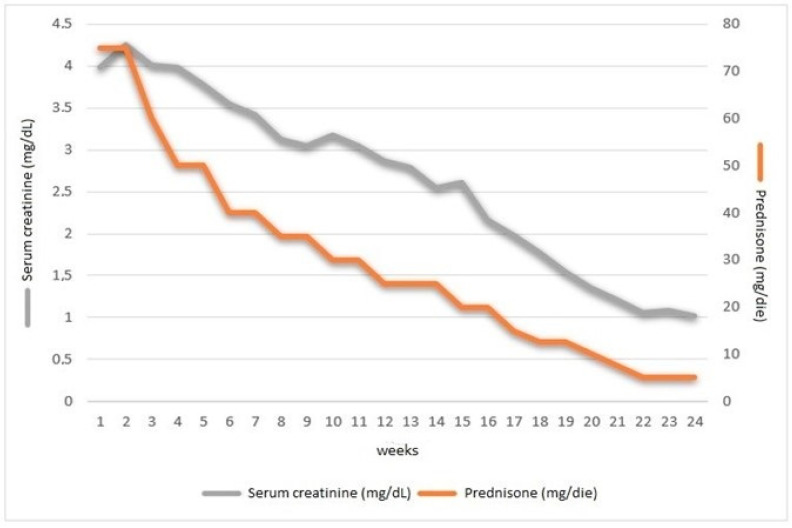
Time course of serum creatinine values during prednisone therapy. Tubulointerstitial nephropathy secondary to *A. lumbricoides* infection was suspected. A percutaneous renal biopsy was not performed since the patient refused the anti-platelet therapy discontinuation, which had been recommended to avoid severe side effects of the biopsy. Antiparasitic therapy with mebendazole (100 mg × 2/day for three days) and albendazole (400 mg once) was administered. Corticosteroid therapy with prednisone (1 mg/kg/day) was also administered for two weeks to improve recovery from renal damage. The dosage was gradually reduced (Figure 3), for a total of 24 weeks of therapy. During clinical–laboratory follow-up, there was the disappearance of the itching and the resolution of the urinary abnormalities. About 3 months later, kidney function significantly increased, although a complete recover to normal was not observed (Figure 3). The patient is currently being followed in our CKD outpatient clinic for residual renal damage, probably associated with the tubular-interstitial fibrosis that rapidly develops following acute kidney injury (AKI). This can be related to the considerable time interval (at least 6 months) between the onset of symptoms and the diagnosis. At 6 months and 1 year after the end of treatment, laboratory tests were stable with normal urine tests, without anaemia and eosinophilia in the blood count, negative stool parasitology tests and slightly increased serum creatinine (1.92 mg/dL with eGFR (CDK-EPI) 37 mL/min/1.73^2^). The patient reported good general condition, without diarrhoea and itching. *Ascaris lumbricoides*, a soil-transmitted large nematode, is the most frequent helminth affecting humans [1]. It causes Ascariasis that can be asymptomatic or symptomatic, with nausea, abdominal pain, bloating and diarrhoea [2]. Ascariasis often progresses insidiously, causing non-specific symptoms. Rarely, acute kidney injury (AKI) occurs mostly as the consequence of an acute interstitial nephritis (AIN) [3], a heterogeneous disorder not only in aetiology but also in presentation, laboratory findings and outcome. AIN is an unusual manifestation among complications caused by *Ascaris*: to date, only three cases have been reported in the literature [4]. Drugs represent the main cause of interstitial nephritis, followed by infectious agents and idiopathic lesions [4]. The aim of this report is to present a rare case of AIN caused by *A. lumbricoides*, one of the most well-known helminthic parasites affecting humans. Despite the existence of many human helminthiases that can trigger kidney damage [5], renal injury in the course of ascariasis rarely occurs [4]. Ascariasis can cause AKI, mostly as the consequence of acute interstitial nephritis (AIN) [6,7], which manifests itself as a form of a hyperergic, hypersensitivity reaction with tubulointerstitial eosinophil infiltration [4,8]. Renal involvement in the context of human parasitosis may have different clinical presentations, spanning from a direct, massive parenchymal invasion to a targeted glomerular or tubulointerstitial involvement [9]. In this clinical case, *A. lumbricoides* was the most likely cause of overt AIN, following AKI, with almost a full recovery made due to a combination of anti-parasitic and corticosteroid therapy. At the time of the onset of symptoms, no drugs had been added that could justify the onset of acute interstitial nephritis disease, and, moreover, the presence of loose stool associated with itching and eosinophilia suggested a parasitic aetiology. Although the diagnosis of AIN remains strongly supported by the clinical manifestations observed, the lack of a histological confirmation represents a key limit. Nevertheless, we do believe that the histological confirmation of AIN would have not provided additional insights into the disease course; neither would it have driven a different therapeutic approach. In fact, the long-term corticosteroid regimen started on an empirical basis was revealed to be efficacious in improving the clinical picture. The improvement in renal function after worm eradication strongly suggested a causal relationship between Ascaris infection and AIN. Although the diagnosis of AIN, as an uncommon complication in *A. lumbricoides* infection, is difficult, parasite infection should be considered in the differential diagnosis of an unexplained decline in renal function. Early diagnosis and treatment are necessary to avoid irreversible complications.

## Data Availability

All data are provided within the manuscript.

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
