# Peer review of "A Rare Complication of Ascariasis: A Case of Acute Interstitial Nephritis"

_diagnostics, 2023, doi:10.3390/diagnostics13122054_

Round 1

Reviewer 1 Report

Abstracts and keywords are acceptable but some keywords need to be amended to match MeSH. 

Congratulations on an article well written. However, there are minor English editing required, for example, lines 43-47 may be improved better. 

Author Response

Comment 1.1: Abstracts and keywords are acceptable but some keywords need to be amended to match MeSH.

Reply 1.1 : We appreciate the Reviewer's comment regarding the modification of some keywords to match the MeSH. Therefore, we changed some keywords.

Comment 1.2: Congratulations on an article well written. However, there are minor English editing required, for example, lines 43-47 may be improved better. 

Reply 1.2 : We thank the Reviewer for the compliments on our manuscript. Based on his suggestions, we have modified lines 43-47 by improving sentence construction in English.

Reviewer 2 Report

Authors should be congratulated for their work. The topic represents a challenging-to-treat differential diagnosis that must be considered in the clinical evaluation of renal failure. The presentation of CR is clear and the manuscript is well-written. The discussion is well-structured. 

Author Response

Comment 2.1: Authors should be congratulated for their work. The topic represents a challenging-to-treat differential diagnosis that must be considered in the clinical evaluation of renal failure. The presentation of CR is clear and the manuscript is well-written. The discussion is well-structured. 

Reply 2.1:

We thank the reviewer for his appreciation of our manuscript.

Reviewer 3 Report

Por insuficiencia renal, un hombre de 72 años ingresó en la Unidad Operativa de Nefrología-Diálisis del Hospital Universitario “Magna Graecia” de Catanzaro, Italia. Presentó 36 con prurito difuso, heces blandas aDebido a insuficiencia renal, un hombre de 72 años ingresó en la Unidad Operativa de Nefrología-Diálisis 35 del Hospital Universitario “Magna Graecia” de Catanzaro, Italia. Presentó 36 con prurito difuso, heces sueltas un
